# Serum and CSF Biomarkers Predict Active Early Cognitive Decline Rather Than Established Cognitive Impairment at the Moment of RRMS Diagnosis

**DOI:** 10.3390/diagnostics12112571

**Published:** 2022-10-23

**Authors:** Vlad Eugen Tiu, Bogdan Ovidiu Popescu, Iulian Ion Enache, Cristina Tiu, Elena Terecoasa, Cristina Aura Panea

**Affiliations:** 1Neurology Department, “Carol Davila” University of Medicine and Pharmacy, 050474 Bucharest, Romania; 2Neurology Department, Elias University Emergency Hospital, 011461 Bucharest, Romania; 3Neurology Department, Colentina Clinical Hospital, 020125 Bucharest, Romania; 4Neurology Department, University Emergency Hospital of Bucharest, 050098 Bucharest, Romania

**Keywords:** cognitive impairment, multiple sclerosis, predictive biomarkers, RRMS, cognitive decline, neurofilaments, amyloid

## Abstract

**(1) Background**: Cognitive impairment (CI) begins early in the evolution of multiple sclerosis (MS) but may only become obvious in the later stages of the disease. Little data is available regarding predictive biomarkers for early, active cognitive decline in relapse remitting MS (RRMS) patients. **(2) Methods**: 50 RRMS patients in the first 6 months following diagnosis were included. The minimum follow-up was one year. Biomarker samples were collected at baseline, 3-, 6- and 12-month follow-up. Cognitive performance was assessed at baseline and 12-month follow-up; **(3) Results**: Statistically significant differences were found for patients undergoing active cognitive decline for sNfL z-scores at baseline and 3 months, CSF NfL baseline values, CSF Aβ42 and the Bremso score as well. The logistic regression model based on these 5 variables was statistically significant, χ2(4) = 22.335, *p* < 0.0001, R2 = 0.671, with a sensitivity of 57.1%, specificity of 97.4%, a positive predictive value of 80% and a negative predictive value of 92.6%. **(4) Conclusions**: Our study shows that serum biomarkers (adjusted sNfL z-scores at baseline and 3 months) and CSF biomarkers (CSF NfL baseline values, CSF Aβ42), combined with a clinical score (BREMSO), can accurately predict an early cognitive decline for RRMS patients at the moment of diagnosis.

## 1. Introduction

Multiple sclerosis (MS) is a chronic, progressive disease of the central nervous system that is characterized by a complex process of neuroinflammation, demyelination and sequential axonal loss (neurodegeneration) [1].

The impact of the disease on cognition has been increasingly recognized and is now considered to be a common feature of MS, affecting 43–70% of patients. While frank dementia is rare, important impairment of essential cognitive domains such as speed of information processing, memory, attention, executive functions and visual perceptual functions may occur [2].

Recent studies have shown that cognitive impairment (CI) begins early in the evolution of the disease but may only become obvious in the later stages [3,4]. Mild cognitive deficits may remain undetected for years due to the complex and time-consuming task of properly evaluating neurocognitive performance in these patients [5].

CI in MS is less frequent or severe in RRMS than in progressive subtypes. SPMS patients seem to be more frequently affected than PPMS subjects, although some authors showed similar degrees of CI between the two forms [6]. However, some RRMS relapses have been proven to significantly alter cognitive performance with various degrees of recovery, leading to the implementation of the term “cognitive relapse” [7]. Previous studies have reported male sex, lower education level and higher EDSS scores to be predictors of CI for RRMS patients [8].

Numerous biomarkers that have been correlated with CI in MS were considered for the design of this study.

Imaging biomarkers were considered. As the paradigm has shifted from considering MS exclusively a white matter (WM) disease and grey matter (GM) pathology became increasingly recognized and quantified [9,10], a surprising finding was that CI in MS showed a better correlation with MRI biomarkers of GM rather than WM pathology [11,12].

While great progress has been made in detecting GM pathological changes on MRI scans, in vivo usage of biomarkers such as cortical and deep gray matter atrophy or T2 lesional volume has proven difficult outside of research laboratories and clinical trials. While less than 25% of MS specialists regularly assess GM atrophy for their patients in the US, with even less usage worldwide [ACTRIMS 2023 Forum survey—unpublished data], the most widely used imagistic biomarkers today are the number of T2 hyperintense WM lesions and the number of Gadolinium enhancing lesions (GdE) [13].

Another commonly available imagining biomarker is optical coherence tomography (OCT). OCT is a non-contact imaging method used to generate high-resolution cross-sectional images of the retina [14]. The thickness of the axonal mesh that creates the retinal nerve fiber layer (RNFL) correlates with pathological changes in MS, namely, disease severity, duration, axonal loss and grey matter atrophy [15]. The RNFL has also been shown to correlate with CI in MS (proving predictive power as a biomarker for cognitive decline) [15,16]. The macular ganglion cell-inner plexiform layer (GCL-IPL) has also been correlated with early cognitive impairment in Alzheimer’s disease [17,18], and with disease duration and severity in MS [19,20].

Many molecules were investigated as potential humoral biomarkers for cognitive impairment (CI) in MS. Of these, we decided to analyze the neurofilament light chains (serum and CSF), CSF Aβ42 fraction, immunoglobulin G index and CSF OCBs for our study due to compelling evidence published in recent years [21,22,23,24,25].

Oligoclonal immunoglobulin G (IgG) bands (OCBs) represent intrathecal IgG synthesis, reflecting the compartmentalized central nervous system (CNS) humoral immune activation present in MS. While not specific to this disease, they are present in over 90% of MS patients and have been used for decades as diagnosis and prognosis biomarkers. The presence of two or more OCBs is considered positive [26,27]. Another biomarker reflecting quantitative IgG intrathecal synthesis is the IgG index, which can be obtained by dividing the IgG CSF plasma concentration quotient by the albumin concentration quotient. A value of 0.7 or more is considered positive, serving as both a diagnosis and predictive biomarker, correlating with early disease activity [28].

The main protein found in senile plaques is called Beta-amyloid (Aβ). Of the two major forms, Aβ42 is of particular interest [29]. Aβ42 was found to be decreased very early in the evolution of Alzheimer’s disease, with numerous studies proving its value as a biomarker for this disease [30,31,32,33]. A range of theories have been proposed as the mechanism behind this phenomenon, including metabolic disturbances, neuronal dysfunction, deposition of insoluble aggregates at the cortical level and even detection issues by ELISA kits [30].

It is still debated what role the amyloid proteins play in MS pathology. Some authors suggest they are involved in the neuropathological process of neuronal degeneration in MS, while others suggest they are rather an epiphenomena of neuroaxonal repair (a third, proper protective role, has been questioned in recent years) [34]. Many studies have shown nonetheless a correlation between low levels of Aβ (mostly Aβ42) in CSF and a poorer prognosis for MS patients. This corelates not only to EDSS but also to cognitive performance on future follow-up [35,36]. Whether the mechanism that links the cognitive decline in MS with lower levels of Aβ42 in CSF is similar to that in Alzheimer’s disease remains to be seen.

Neurofilaments (Nf) are physiologic component proteins of the axonal cytoskeleton. They play a role in axonal stability, radial growth and caliber maintenance, as well as electrical-impulse transmission [37]. Nf are composed of the following four subunits: heavy (NfH), median (NfM) and light (NfL) chains and α-internexin (Int). The most studied of these subunits, being the most abundant and soluble, is the light chain (NfL) [38]. Nf are constantly released in the CSF (and eventually into the blood) under normal conditions (weight, sex and age play important roles in what amount is considered within the normal range). Many studies have shown their levels rise with axonal damage, making them excellent biomarkers of disease activity [39]. Axonal damage and neuronal loss are present from the earliest stages of MS and are linked with cortical atrophy, fatigue, progression of disability and cognitive dysfunction [40]. Numerous studies in the past decade have proven a link between elevated NfL levels (in both CSF and serum) and cognitive impairment in MS [41,42].

A group led by Prof. Kuhle recently published a tool for modeling the distribution of sNfL concentration adjusted according to age and BMI and deriving a percentile and z-score value using a healthy control reference database. Percentiles and z-scores showed good predictive power for MS patients, being correlated to a gradually increased risk of future acute and chronic disease activity. The Z-scores for sNfL outperformed raw values for diagnostic accuracy [43].

Many clinical scores have been proposed along the way to predict unfavorable outcomes in MS. The RoAD score calculates the 10-year risk of ambulatory disability (EDSS = 6.0) after 1-year of follow-up under DMT (based on sex, age, duration of symptoms, baseline EDSS, EDSS progression, number of relapses in the first year of DMT and new T2 lesions on follow-up MRI) [44].

The Bayesian risk estimate for MS at onset (BREMSO) score was designed to predict unfavorable long-term evolution for MS patients (defined as expanded disability status scale, EDSS, ≥6.0 and onset of the secondary progressive (SP) phase of the disease). The Bayesian risk estimate for MS at onset (BREMSO) [45] score was designed to predict unfavorable long-term evolution for MS patients (defined as expanded disability status scale, EDSS, ≥6.0 and onset of the secondary progressive (SP) phase of the disease). The score is based on data available at baseline assessment (age, sex, sphincter onset, motor onset, motor and sensory onset, number of functional systems involved at onset, incomplete recovery after onset).

Since most of these metrics used in the two scores have been previously tied to cognitive impairment on long-term follow-up, we decided to include them in our analysis.

It is worth mentioning that the mechanisms responsible for CI in MS are not completely understood. Complex biological mechanisms involving both immunological and neurodegenerative processes that lie at the base of MS pathology have been associated with CI [46], and thus we have to consider that structural injury is intertwined with the functional impairment of neuronal networks in producing cognitive dysfunction [47].

Cognitive impairment, fatigue and depression often coexist in MS patients. Depression has a lifetime prevalence of approximately 50% and an annual prevalence of around 20% in MS patients, or around 2–3 times higher than the general population [48,49,50]. Depressive disorders may also lead to various degrees of cognitive impairment, varying in impact with patient age and depression severity (more severe symptoms being associated with worse cognitive performance) [51,52]. When evaluating cognitive performance in MS patients, depression must always be assessed as well.

We decided to test the following two hypotheses for patients that have just been diagnosed with RRMS: first, do the easily accessible, widely used biomarkers of today correlate with cognitive parameters early in the course of the disease, and second, can these biomarkers predict poor cognitive outcomes early in the course of the disease?

## 2. Materials and Methods

We enrolled 50 consecutive patients that had been newly diagnosed with relapsing-remitting multiple sclerosis (RRMS) in our Center (Neurology Department of the Emergency University Hospital of Bucharest). Enrollment began in June 2020 and ended in June 2021.

The inclusion criteria were as follows: diagnosis of MS according to McDonald Criteria 2017 [53] that has been established no longer than 6 months prior, age ≥ 18 years old, signed informed consent for study participation. Exclusion criteria were as follows: associated medical conditions that would impair evaluation or prevent participation in study protocol—severe Psychiatric or medical conditions that would prevent clinical evaluation or collaboration, history of alcohol or drug abuse, pregnancy.

We collected clinical-demographical data such as gender, age, time from onset of symptoms to diagnosis, race, level of education, lifestyle, clinical scores such as expanded disability status score (EDSS), Montreal cognitive assessment (MoCA), oral symbol digit modalities test (SDMT), as well as the brief visuospatial memory test-revised (BVMT-R). All patients were screened periodically (inclusion, 6, 12 months) for depression using the BDI-II scale.

All the patients underwent a baseline contrast enhanced cerebral MRI and a follow-up scan at 1-year. Biological samples were collected at baseline (CSF, serum), and at 3- and 6-months follow-up (serum). Corticosteroids or other MS-specific immune-active therapies were initiated after inclusion and baseline sample collection, when appropriate. No subjects were taking psychoactive drugs or substances that might interfere with neuropsychological performance.

All OCT assessments were performed in the same center, on a CIRRUS™ HD-OCT 5000 machine, by expert ophthalmologists. Metrics obtained from eyes with a history of optic neuritis were excluded from the final analysis.

All patients were followed-up for a minimum of 1-year.

Cognitive assessment was performed at baseline and 1-year follow-up using MoCA, oral SDMT and BVMT-R scores.

### 2.1. CSF and Serum Analysis, Biomarkers Determination

Serum samples were collected by peripheral vein punction. Samples were left to clot for 40 min, then centrifugation was performed for 10 min at 2000 rotations per minute. After that, the samples were aliquoted in polypropylene tubes and stored at −80 degrees Celsius. CSF samples were collected at inclusion by lumbar puncture. After centrifugation for 10 min at 2000 rotations per minute, the samples were aliquoted in polypropylene tubes and stored at −80 degrees Celsius.

All patients were tested for cell count, glucose and protein CSF, as well as oligoclonal bands, IgG index, CSF neurofilaments and beta amyloid (Aβ42 fraction). Serum samples were sent to an external lab for neurofilament light chain detection using SIMOA assays method.

CSF and serum samples were analyzed by certified laboratory technicians, blinded to clinical data.

### 2.2. Data Availability and Statistical Analysis

All the participants gave their written consent to participate in the study and for the storage of biological samples for research purposes. The study was given approval by the ethics committee of the University Emergency Hospital of Bucharest. Collected data were used to produce a pseudonymized dataset, available under reasonable request to the corresponding author. Statistical analysis was performed using SPSS 26.0 for Windows (SPSS Inc., Chicago, IL, USA) and Microsoft Excel.

We checked the normality distribution of data with the Kolmogorov-Smirnov Test and Shapiro-Wilk Test. We presented categorical data with median, proportions as numbers and percentages. Mean and standard deviation was used for continuous data. Independent T^2^ test, Mann–Whitney U test and Kruskal–Wallis test were used for comparison between continuous variables. The chi-squared test and Fisher test were used for categorical variables. Bonferroni correction was performed for multiple comparison analysis, and Spearman’s rank correlation coefficient test was used for the correlation between continuous variables. Linear regression and binary logistic regressions were performed to establish predictive models between cognitive performance (MOCA, SDMT, BVMT-R, cognitive impairment, cognitive decline over 1-year follow-up) and measured biomarkers, either as continuous variables (sNfL samples at baseline, at the 3 and 6 months follow-up, CSF NfL at baseline, CSF Aβ42, OCT metrics, BREMSO and RoAD scores) or dichotomous variables (sex, education level, clinical depression, MRI parameters-20 T2 hyperintense lesions or more on baseline MRI, 2 Gadolinium Enhancing Lesions (GDE) or more on baseline MRI, 2 or more T2 hyperintense lesions or 1 or more GDE lesion on the 1-year follow-up MRI, cognitive impairment at baseline and 1-year follow-up, altered cognitive performance at 1-year follow-up).

All tests were two-sided and the significance threshold was set to *p* < 0.05 unless otherwise specified.

## 3. Results

Our lot included a total of 50 patients that completed the 1-year follow-up while fully respecting the study protocol. Of these, 35 patients were women (70%) and 15 were men (30%). The mean age was 2.6 years at inclusion (18; 52). Table 1 shows the main characteristics of the patient lot.

We decided to test if there was any significant statistical difference in cognitive performance between patients depending on their BDI II scores across all cognitive tests performed. The data can be found in Appendix A. No statistically significant difference was found in cognitive performance between the groups for baseline testing and 1-year follow-up.

We also analyzed to see if patients that had an increase in BDI II score between the first evaluation and the 1-year follow-up showed any correlation with changes in their cognitive performances. We ran Pearson correlations for both changes in BDI II scores and MoCA, or SDMT, respectively. Appendix A shows our results. No statistically significant correlation was found. Following this analysis, we decided that the BDI II score should not be included as a variable in further models.

We then analyzed the impact of the level of education on cognitive testing. No differences were found between the following 2 analyzed groups: high school diploma (12 years of formal education) and college degree (14–16 years of formal education), respectively. Appendix A summarizes our findings.

No statistically significant difference can be outlined between the two groups for baseline testing.

For the 1-year follow-up testing, patients with more years of formal education (group 2) tend to show higher scores on the SDMT test when compared to group 1 (58.4 vs. 50.9 points on average, or an absolute increase of 7.5 points, *p* value = 0.04). A similar tendency is seen for the MoCA testing as well (28 vs. 26.9 points, or a 1.1 point absolute increase); however, the difference is not statistically significant for this test, with a *p* value of 0.18.

### 3.1. Impact of Age on Cognitive Performance

We tested for correlations between the patient’s age and cognitive performance, in both baseline and 1-year follow-up testing. Table 2 includes all correlations that showed statistical significance. All other cognitive tests performed showed a tendency towards a negative correlation with the patient’s age, but *p* values were not statistically significant.

A low-to-moderate negative correlation was found for oral SDMT at 1-year follow-up, baseline MoCA and BVMT-R evaluations. Age will therefore be considered a covariable when further analyzing these tests.

### 3.2. Correlations between Clinical Impact of MS and Cognitive Performance

No statistically significant correlation was found between either baseline or 1-year follow-up cognitive testing (MoCA, SDMT, BVMT-R) and the clinical impact of MS as assessed by the EDSS scale.

### 3.3. Effect of DMTs on Biomarkers and Cognitive Performance

Only 6 patients were started directly on high-efficacy DMTs following RRMS diagnosis (either Natalizumab or Ocrelizumab), while 44 were initiated on first-line agents. The median follow-up under any DMT for the study population was 10 months. The last blood sample was obtained when patients had been under DMTs for a median time of 4 months. We compared all biomarkers and cognitive tests at 1-year follow-up between the two groups, but no statistically significant difference was found. DMTs showed no impact on predictive modeling in further analysis. Amongst the patients treated with first-line agents, no significant differences were found between any of the used agents regarding cognitive performance or analyzed biomarkers. We decided not to include this variable in further analysis.

### 3.4. MoCA Score

Baseline score

Baseline MoCA scores were not statistically significantly correlated with any of the predictive biomarkers we analyzed in our study.

One-year follow-up

Table 3 summarizes the key biomarkers that showed a statistically significant correlation with the one-year MoCA test scores. 

A moderate negative correlation was found between 1-year follow-up MoCA scores and ROAD scores, CSF NfL baseline values, raw sNfL values at baseline, 3- and 6-months follow-up, as well as sNfL z-scores at 3 months follow-up.

Multilinear regression model

Despite the fact that some variables have shown an initial correlation with baseline and 1-year MoCa scores, no predictive factor were determined by the regression analysis.

### 3.5. SDMT Score

Baseline score

Baseline SDMT scores were not statistically significantly correlated with any of the predictive biomarkers we analyzed in our study.

One-year follow-up

Table 4 summarizes the key biomarkers that showed a correlation with the 1-year SDMT test score. 

A moderate negative correlation was found between the 1-year follow-up oral SDMT scores and the RoAD score, 6 months follow-up sNfL raw values and the average thickness of the GCL-IPL layer.

Multilinear regression analysis

We decided to run a linear regression model using four variables that showed a linear correlation with SDMT scores at one year—age, average GCL IPL layer thickness and sNFL raw values at 6 months follow-up, while using education level as a dichotomous variable.

Two outliers were removed, after which no influential or leverage point adjustments were required.

The multiple regression model statistically significantly predicted the SDMT score, F(3,39) = 4.694, *p* = 0.003, R^2^ = 0.325. All independent variables showed statistically significant power, with the exception of education level. We decided the variable should be kept within the model due to a strong B coefficient of 4.48. Table 5 summarizes our key findings.

### 3.6. BVMT-R

Baseline score

Baseline BVMT-R scores were not correlated with any of the predictive biomarkers we analyzed in the study.

One-year follow-up

Table 6 summarizes the key biomarkers that showed a statistically significant linear correlation with the 1-year BVMT-R test scores. 

A weak-to-moderate negative correlation was seen with sNfL raw values at 6 month follow-up across all tests. Isolated weak to moderate negative correlations were found between the BVMT-R 1 test round and RoAD score, as well as the T2 hyperintense lesion burden. A moderate negative correlation was seen for test round 3 with CSF NfL raw values at baseline.

Multilinear regression model

We decided to build our model for BVMTR total score (T1-T3) based on age and raw sNfL values at 6 months follow-up and the presence of 20 or more T2 hyperintense lesions on the baseline MRI.

Four outliers were excluded from the analysis, after which no further influential or leverage point adjustments were required.

The multiple regression model statistically significantly predicted the BVMTR total score at 1-year follow-up, f (3.46) = 3.086, *p* = 0.036, R^2^ = 0.168. Notably, none of the independent variables showed statistically significant power. Table 7 includes our most relevant findings.

### 3.7. Cognitive Impairment at RRMS Diagnosis and 1-Year Follow-Up

We defined cognitive impairment (CI) within our group as either a MoCA score of fewer than 26 points or an SDMT oral score of fewer than 35 points. A total of 20 patients (40%) of our cohort matched these criteria at baseline testing, while only 7 patients (14%) had CI at 1-year follow-up.

At both baseline and 1-year follow-up, the two groups (cognitive impairments vs. no cognitive impairment) showed no statistically significant difference between any of the analyzed biomarkers or clinical status, as reflected by the EDSS scale (tested by Fisher exact test and independent samples T-test).

### 3.8. Multiple Linear Regression Models for Predicting Cognitive Impairment at RRMS Diagnosis and 1-Year Follow-Up

We analyzed our data to see if there was any trend for differences in group means for the selected biomarkers between the two groups (cognitive impairment vs. no cognitive impairment). No statistically significant differences were found.

No predictive factor was determined by the regression analysis.

### 3.9. Cognitive Decline in the First Year following Diagnosis

We defined cognitive decline in the first year of follow-up as assessed by either a decrease in MoCA score of two or more or a decrease in SDMT score of four or more points. A total of nine patients met the criteria (Note: this analysis differs from the one above regarding established cognitive impairment. For this analysis, we reanalyzed the whole group related to a drop in cognitive performance as described, whether or not this led to meeting cognitive impairment criteria at the 1-year follow-up or not).

### 3.10. Predictive Modelling for Cognitive Decline in the First Year following Diagnosis

We decided to run a binomial logistic regression model for cognitive decline at a 1-year follow-up. We ran multiple possible models, selecting variables based on previous literature reports and on our own data. Information regarding our statistical process can be found in Appendix A. We finally decided to use sNfL z-scores at baseline and 3 months, CSF NfL baseline values, CSF Aβ42 and the Bremso score as well, leading to a model consisting of five variables.

The linearity of the continuous variables with respect to the logit of the dependent variable was assessed via the Box–Tidwell (1962) procedure. A Bonferroni correction was applied using all 10 terms in the model, resulting in statistical significance being accepted when *p* < 0.0005 (Tabachnick & Fidell, 2014). Based on this assessment, all continuous independent variables were found to be linearly related to the logit of the dependent variable.

There were four outliers with values greater than three standard deviations, which were removed. The final analysis contained 39 patients with no cognitive decline and 7 patients with cognitive decline.

The area under the ROC curve was 0.952 (95% CI 0.887 to 1.0), *p* > 0.0001, with an excellent level of discrimination, as seen below in Figure 1.

The logistic regression model was statistically significant, χ2(4) = 22.335, *p* < 0.0001. The model explained 67.1% (Nagelkerke R^2^) of the variance in cognitive decline and correctly classified 91.3% of cases. Sensitivity was 57.1%, specificity was 97.4%, positive predictive value was 80% and negative predictive value was 92.6%.

## 4. Discussion

We recruited a cohort of 50 consecutive patients that had been recently diagnosed with RRMS. Of these, 20 patients (40%) fulfilled the criteria for cognitive impairment at baseline assessment. No biomarkers or clinical scores showed a statistically significant difference between the two groups (cognitive impairment vs. no cognitive impairment) initially. When the same population was tested again at 1-year follow-up, only 7 patients (14%) still met the criteria for CI, but there was once again no difference between the 2 groups for the measured biomarkers.

This may be due to multiple reasons, perhaps the most important of which is the practice effect reported for both MoCA and SDMT scores [54,55]. Other reasons could include the DMT effect and improved performance after adjusting to the setting of a clinical trial. The large variation in the percentage of the population affected by cognitive impairment is not unusual for MS patients and may explain the large reference intervals also seen in the literature.

No correlations were found between BDI II scores and cognitive performance, although notably, our lot had no patients with severe forms of depression.

No statistically significant difference was found between lower and higher education patient groups regarding cognitive performance on baseline testing. Improvements were seen on repeat testing at 1-year follow-up for the higher education group, mostly however with no statistical significance. The notable exception was the SDMT score, which improved on average by 7.5 points (*p* = 0.04) for the higher education group. MoCA testing showed a similar trajectory with an improvement of 1.1 points, although this did not reach statistical significance (*p* = 0.18). As previously mentioned, we believe this to be a result of the practice effect, where patients with more years of formal training appear to be more susceptible to performance improvement on repeat testing.

Age was negatively correlated with cognitive performance, younger patients performing better on baseline MoCA and BVMTR, and 1-year follow-up SDMT scores. This phenomenon has been reported by previous studies as well for MS patients [56], and age was included as a covariable in our analysis where appropriate.

No correlation was found between the clinical impact of MS, as assessed by EDSS score, and cognitive impairment. While many studies have shown a correlation between EDSS and cognitive decline on long-term follow-up [57,58], our data draw attention that this is not the case for baseline assessment and early disease evolution.

The MoCA score did not correlate in a statistically significant manner with any of the analyzed biomarkers on baseline testing. While some biomarkers showed weak correlations with the 1-year follow-up MoCA score, no predictive factors were determined by the regression analysis.

SDMT (oral) testing also did not show any correlations at baseline testing with any of our biomarkers. For the 1-year follow-up scores, weak negative correlations were found with the RoAD score, raw values of sNfL at 6 months follow-up and the average thickness of the GCL-IPL layer. We constructed a prediction model based on age, education level, sNFL values at 6 months follow-up and GCL-IPL layer average thickness, with an R^2^ of 0.342. We consider this an important finding, showing that NfL values at 6 months (rather than baseline or 3 months) and GCL-IPL layer thickness are predictive biomarkers for early cognitive impairment as measured by SDMT testing.

BVMTR testing at baseline showed no correlation with any of the biomarkers we analyzed. One-year follow-up testing showed correlations only for the 6 months follow-up raw sNFL values, but scores from the first round of testing also showed a statistically significant difference between groups on the number of T2 hyperintense lesions at baseline. Patients with 20 or more such lesions on baseline MRI showed, on average, lower BVMTR scores than those that did not. A predictive model was built based on age, raw sNfL values at 6 months follow-up and the presence of 20 or more T2 hyperintense lesions on baseline MRI. The multiple regression model statistically significantly predicted the BVMTR total score at 1-year follow-up, f (3,46) = 3.086, *p* = 0.036, R^2^ = 0.168, albeit the predictive power was poor and none of the variables included showed individual statistical significance.

Perhaps the most relevant part of our study was the analysis of the patients that showed a significant cognitive decline between baseline testing and 1-year follow-up. When comparing the two groups, statistically significant differences were found for sNfL z-scores at baseline and three months, CSF NfL baseline values, CSF Aβ42 and the BREMSO score as well.

The logistic regression model based on these 5 variables was statistically significant, χ2(4) = 22.335, *p* < 0.0001, R^2^ = 0.671, with a sensitivity of 57.1%, specificity of 97.4%, a positive predictive value of 80% and a negative predictive value of 92.6%.

This data shows that serum (adjusted sNfL z-scores at baseline and 3 months) and CSF biomarkers (CSF NfL baseline values, CSF Aβ42), combined with a clinical score (BREMSO), can accurately predict an early cognitive decline for RRMS patients at the moment of diagnosis.

## 5. Conclusions

Our findings show that easily available biomarkers that are widely used today for MS patients linearly correlate and show predictive power for early cognitive performance scores. While patients with already established cognitive impairment showed no relevant differences in the analyzed biomarkers of our study, patients undergoing an active cognitive decline in the first year following RRMS diagnosis had significant changes in both serum and CSF biomarkers.

Unsurprisingly, our analysis showed that sNfL z-scores at baseline and 3 months, rather than raw values, held better predictive power for active cognitive decline. CSF NfL, Aβ42 and a vast array of clinical data, as reflected in the BREMSO score, completed our prediction model.

Our study has a number of drawbacks, among which the most important are the small sample size (50 patients), the monocentric design and the short follow-up time at the time of publishing this article.

The timely detection of RRMS patients that are at high risk for early cognitive decline is of paramount importance, as a quick and aggressive intervention may change the course of the disease. We consider it a strong point for our prediction model that all the input needed is available after just 3 months of follow-up.

Our study underlines that these biomarkers reflect active cognitive decline rather than established cognitive impairment in MS patients and further research is needed to investigate this, perhaps in the setting of larger clinical trials.

## Figures and Tables

**Figure 1 diagnostics-12-02571-f001:**
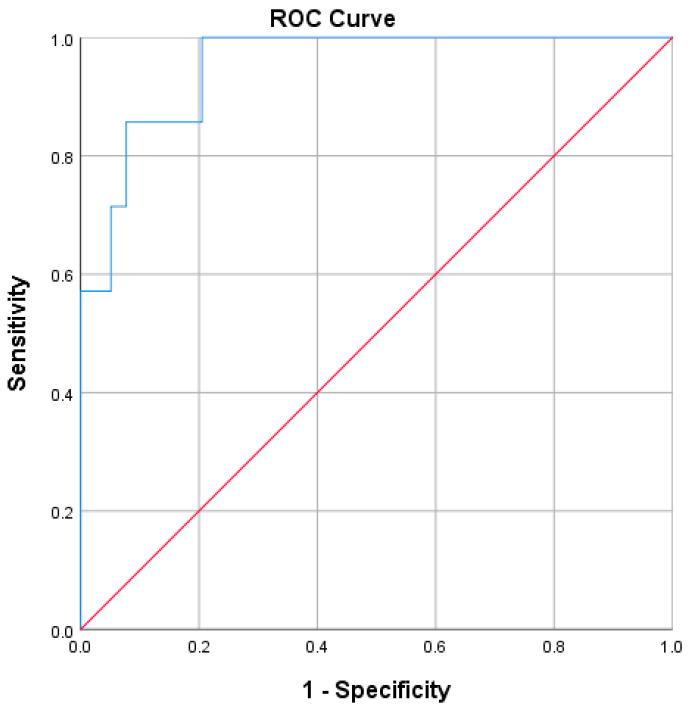
ROC curve of the predictive model for cognitive decline in the first year following diagnosis of RRMS.

**Table 1 diagnostics-12-02571-t001:** Shows the main characteristics of the lot at the time of study inclusion and the data available after 1-year follow-up.

	Median (Minimum; Maximum Values)
**Age**	29.6 (18; 52)
**Female/male**	35 F (70%)/15 M (30%)
**EDSS**—baseline	1. (0; 6)
**EDSS**—1-year follow-up	1.8 (0; 6.5)
**Considered to be aggressive MS by attending physician**	6 (12%)
**Relapses in the first year** 0 relapses 1 relapse 2 relapses	33 (66%) 16 (30%) 2 (4%)
**Classified as active MS at 1-year follow-up**	23 (45.1%)
**CSF immunological analysis**	**No. of patients (%)**
Positive oligoclonal bands	36 (72%)
Positive immunoglobulin index	12 (24%)
**Baseline MRI characteristics**	
20 or more T2/FLAIR hyperintense lesions	35 (70%)
2 or more Gadolinium enhancing lesions (GdE)	15 (30%)
**1-year follow-up MRI characteristics**	
2 or more new T2/FLAIR hyperintense lesions	22 (44%)
One or more new GdE	7 (14%)
**Baseline OCT characteristics**	**Mean (minimum; maximum values)**
RNFL	95.3 (73; 118)
GCL + IPL	78.6 (55.5; 92.5)
**Neurofilaments**	
Baseline sNfL raw values	20.5 (3.2; 208)
Baseline sNfL adjusted z-score	1.8 (−2.27; 3.81)
Baseline CSF NfL	2044.2 (201; 16,857)
3 months follow-up sNfL raw values	12.7 (2.9; 49.8)
3 months follow-up sNfL adjusted z-score	1.4 (−2; 3.35)
6 months follow-up sNfL raw values	10.5 (2.77; 31.7)
6 months follow-up sNfL adjusted z-score	1.1 (−1.8; 3.16)
**CSF Beta-amyloid**	671.3 (280; 1322)
**Predictive scores**	
BREMSO	0.4 (−0.65; 2.39)
RoAD (baseline score)	2.3 (0.5)
RoAD (1-year follow-up final score)	3.2 (0.7)
**Years of education**	No. of patients (%)
Highschool diploma (12 years)	20 (40%)
College or more (14–16 years)	30 (60%)

BREMSO: Bayesian risk estimate for MS at onset; CSF: corticospinal fluid; EDSS: expanded disability status score; GCL + IPL: ganglion cell inner plexiform layer; MRI: magnetic resonance imaging; MS: multiple sclerosis; OCT: optical coherence tomography; RNFL: retinal neve fiber layer; RoAD: risk of ambulatory disability; sNFL: serum neurofilaments.

**Table 2 diagnostics-12-02571-t002:** Correlations between cognitive performance and age.

	R Value	*p*-Value
SDMT 1-year follow-up	−0.414	0.003
MoCA	−0.33	0.019
Baseline BVMT-R (1,2,3, Total score and DR)	−0.35/−0.29/−0.388/−0.378/−0.335	0.012/0.035/0.005/0.007/0.017

BVMT-R: brief visuospatial memory test-revised; BVMTR-DR: brief visuospatial memory test-revised—delayed recall; MoCA: Montreal cognitive assessment; SDMT: symbol digit modalities test.

**Table 3 diagnostics-12-02571-t003:** Biomarker Correlations with One-Year Follow-up MoCA Test Score.

Biomarker	R Value	*p*-Value
RoAD score	−0.34	0.015
sNfL baseline raw values	−0.33	0.019
sNfl raw values—3 months follow-up	−0.32	0.021
sNfl z-score—3 months follow-up	−0.32	0.022
CSF NfL raw values	−0.30	0.033
sNfl raw values—6 months follow-up	−0.42	>0.001

CSF: corticospinal fluid; RoAD score: risk of ambulatory disability score; NfL: neurofilaments; sNFL: serum neurofilaments.

**Table 4 diagnostics-12-02571-t004:** Biomarkers Correlations with One-Year Follow-up SDMT Test Score.

Biomarker	R Value	*p*-Value
RoAD score	−0.35	0.012
sNfl raw values—6 months follow-up	−0.36	0.01
Average GCL-IPL thickness	0.30	0.035

GCL + IPL: ganglion cell inner plexiform layer; RoAD score: risk of ambulatory disability score; sNFL: serum neurofilaments.

**Table 5 diagnostics-12-02571-t005:** Multiple regression predictive model for 1-year follow-up SDMT scores.

SDMT One-Year Follow-Up	B	95% CI for B		SE B	β	R^2^	ΔR^2^
		LL	UL				
Model						0.342	0.295
Constant	43.5 **	12.12	74.97	15.57			
Age	−0.558 **	−0.940	−0.176	0.189	−0.370 **		
GCL-IPL thickness	0.437 *	0.063	0.810	0.185	0.297 *		
Raw sNFL values at 6 months	−0.561 **	−0.985	−0.136	0.210	−0.335 **		
Education level	4.480	−2.534	11.495	3.468	0.185		

Note. Model = ”Enter” method in SPSS Statistics; B = unstandardized regression coefficient; CI = confidence interval; LL = lower limit; IL = upper limit; SE B = standard error of the coefficient; β = standardized coefficient; R^2^ = coefficient of determination; ΔR^2^ = adjusted R^2^. * *p* < 0.5, ** *p* < 0.1. GCL + IPL: ganglion cell inner plexiform layer; SDMT: symbol digit modalities test; sNFL: serum neurofilaments.

**Table 6 diagnostics-12-02571-t006:** Biomarker Correlations with the One-Year Follow-up BVMT-R Tests Score.

**BVMT-R 1 1-year follow-up**		
**Biomarker**	**R value**	***p*-value**
RoAD score	−0.288	0.042
sNfL raw values 6 months follow-up	−0.29	0.04
More than 20 T2 lesions on baseline MRI	−0.280	0.04
**BVMT-R 2 1-year follow-up**		
**Biomarker**	**R value**	***p*-value**
sNfL raw values 6 months follow-up	−0.307	0.03
**BVMT-R 3 1-year follow-up**		
**Biomarker**	**R value**	***p*-value**
sNfL raw values 6 months follow-up	−0.296	0.037
CSF NfL raw values at baseline	−0.306	0.034
**BVMT-R total score T1-T3 1-year follow-up**		
sNfL raw values 6 months follow-up	−0.289	0.042
**BVMT-R DR 1-year follow-up**		
**Biomarker**	**R value**	***p*-value**
sNfL raw values 6 months follow-up	−0.286	0.049

BVMT-R: brief visuospatial memory test-revised; BVMTR-DR: brief visuospatial memory test-revised—delayed recall; CSF: cerebrospinal fluid; RoAD score: risk of ambulatory disability score; MRI: magnetic resonance imaging; NfL: neurofilaments; sNfL: serum neurofilaments.

**Table 7 diagnostics-12-02571-t007:** Multiple regression results for 1-year follow-up BVMTR total score.

BVMTR 1-Year Follow-Up Total Score	B	95% CI for B		SE B	β	R^2^	ΔR^2^
		LL	UL				
Model						0.168	0.113
Constant	36.457 ***	28.452	44.462	3.977			
Age	−0.162	−0.404	0.080	0.120	−0.184		
Raw sNfL values at 6 months	−0.281	−0.568	0.006	0.143	−0.266		
More than 20 T2 lesions on baseline MRI	−3.119	−7.559	1.321	2.206	−0.194		

Note. Model = ”Enter” method in SPSS Statistics; B = unstandardized regression coefficient; CI = confidence interval; LL = lower limit; IL = upper limit; SE B = standard error of the coefficient; β = standardized coefficient; R^2^ = coefficient of determination; ΔR^2^ = adjusted R^2^. *** *p* < 0.001. BVMT-R: brief visuospatial memory test-revised; MRI: magnetic resonance imaging; sNfL: serum neurofilaments.

## Data Availability

Collected data were used to produce a pseudonymized dataset, available under reasonable request to the corresponding author.

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
