# Peer review of "Serum and CSF Biomarkers Predict Active Early Cognitive Decline Rather Than Established Cognitive Impairment at the Moment of RRMS Diagnosis"

_diagnostics, 2022, doi:10.3390/diagnostics12112571_

Round 1
Reviewer 1 Report
The current paper is a small study that corroborates previous studies that Nfl is a significant predictor of cognitive decline in patients with RRMS especially if they already had detectible cognitive decline at the start of the study.
Author Response
To our esteemed reviewer,
Thank you for taking the time to offer your expertise on our submitted manuscript. We hope that our work will open the way to larger, longer clinical studies investigating active cognitive decline in MS and its biomarkers.
Reviewer 2 Report
The presend manuscript is interesting but the sample size is too small and the follow-up time too short . Due to these limitations the researchers couldn't lead to reliable conclusions.
I reccomend the authors to improve the specific limitations and submit their new results
Author Response
To our esteemed reviewer,
We thank you for your valuable feedback, and for taking the time to offer your expertise regarding our submitted manuscript.
We agree that the sample size of our study is small, but the population lot is well balanced, and representative of real-life MS patients. Furthermore, we are not investigating novel biomarkers, but rather well-established biomarkers that have been proven to be reliably correlated with cognitive decline in MS. In other words, we are not trying to prove the correlation of the studied biomarkers with cognitive decline in MS, but rather to find correlations between these biomarkers, and which ones may best guide the clinician. Statistical data shows we have included sufficient final events for a reliable analysis.
The short follow-up is due to the design of the study and the hypothesis of our research: can active cognitive decline be predicted in MS patients at the moment of RRMS diagnosis? As proof, active cognitive decline was detected in a key number of our patients within the first year following diagnosis, therefore addressing the question at stake and opening the way for larger, longer follow-up studies to answer broader questions, such as how this translated for long-term (5, 10 years) follow-up, or whether these patients do indeed carry a worse long-term cognitive prognosis than the rest.
To our knowledge, very little information has been published regarding active, early (moment of diagnosis) cognitive decline in MS and related biomarkers. Our findings suggest this can be easily predicted using commonly used tools, and perhaps influence therapeutic decisions for these patients. It is for this reason we have decided to submit our paper despite the shortcomings you mentioned in your review.
We hope that we have provided a valid reason to overlook beyond these limitations, and reconsider the need for publishing these important data at this particular point in time.
Best regards
Reviewer 3 Report
The manuscript addresses a definitely very interesting topic with a potentially great meaning for the modern neurology. However, there are serious issues associated with the study design:
- - The study group is very small and heterogenous and the subgroups with cognitive impairment and/or cognitive decline are too small to build clinically reliable predictive models
- - The influence of disease modifying therapy (DMT) is completely omitted in the study, although the results in this particular group of patients are with no doubt influenced by the introduction of DMT. The short comment in the discussion is unfortunately not enough to correct this issue.
- - The use of MoCA in the study at the very early stage of the disease seems to be not optimal
In general, the study needs to be significantly improved before publication
Author Response
To our esteemed reviewer,
We thank you for your valuable feedback, and for taking the time to offer your expertise regarding our submitted manuscript. We agree that the sample size of our study is small and heterogenous, but the population lot is well balanced and representative of real-life MS patients, this being a cohort study.
We would like to underline that we are not investigating novel biomarkers, but rather well-established biomarkers that have been proven to be reliably correlated with cognitive decline in MS. In other words, we are not trying to prove the correlation of the studied biomarkers with cognitive decline in MS, but rather to find correlations between these biomarkers, and which ones may best guide the clinician. Statistical data shows we have included sufficient final events for a reliable analysis.
We agree that these concerns should be addressed more broadly in the discussion section of our article, and we have added a paragraph accordingly.
There was much debate within our group which cognitive screening tool should be used in the design of our study. We would like to stress that the shared vision on our research was that the tools we investigated should be easily available and reliably performed by most clinicians around the world. For this reason, we chose not to include the more complex data we analysed in the final manuscript (MRI volumetrics, angio-OCT), due to the fact that they are usually unavailable to general MS centres. Therefore, after an extensive literature review, we decided that the quickest and most reliable test would be MOCA rather than BICAMS (we performed other cognitive assessments on our lot as well, including complex gamified tests, BVMTR, SDMT etc.), but 1) they performed poorer than the MOCA scale on statistical analysis and 2) did not meet our criteria for ease of use and access. Numerous articles validate our point of view regarding the sensibility and specificity of MOcA scale for early cognitive impairment in MS patients, we are citing 3 of them below.
- Freitas S, Batista S, Afonso AC, Simões MR, de Sousa L, Cunha L, Santana I. The Montreal Cognitive Assessment (MoCA) as a screening test for cognitive dysfunction in multiple sclerosis. Appl Neuropsychol Adult. 2018 Jan-Feb;25(1):57-70. doi: 10.1080/23279095.2016.1243108. Epub 2016 Oct 28. PMID: 27791389.
- Charvet LE, Taub E, Cersosimo B, Rosicki C, Krupp LB (2015) The Montreal Cognitive Assessment (MoCA) in Multiple Sclerosis: Relation to Clinical Features. J Mult Scler 2:135.
- Dagenais, E., Rouleau, I., Demers, M., Jobin, C., Roger, É, Chamelian, L., & Duquette, P. (2013). Value of the MoCA Test as a Screening Instrument in Multiple Sclerosis. Canadian Journal of Neurological Sciences / Journal Canadien Des Sciences Neurologiques, 40(3), 410-415. doi:10.1017/S0317167100014384
The influence of DMTs was considered in our study, and the database not only followed which dmt the patients were put on, but also if any other acute medication was prescribed (e.g. cortisone), days since last dose before blood samples, no of weeks of administration, etc. We agree that not enough was commented on this. The manuscript was adjusted to reflect your observation. In short, we considered the following: most patients were on DMTs for less than 9 months by the end of the follow-up period. This is much too short to consider it might impact cognitive performance, however we ran statistical tests to ensure this was not the case and no differences were observed between cognitive performance for high efficacy DMT vs first line agents, nor regarding treatment duration etc. Since most molecular biomarkers were collected at baseline, 3 and 6 months following inclusion, when most patients had been on DMTs for an average of 3 months, we also considered that the impact of DMTs on biomarkers such as sNfL was not considerable. Statistical analysis confirmed this. We agree this information is important and have added a paragraph addressing this issue in our manuscript.
Best regards
Reviewer 4 Report
Both the research of cognitive impairment and of biomarkers (fluid and radiological) became hot topics in MS research during the last couple of years. Thus, the topic of the article is both relevant and interesting.
However, there are some issues that should be corrected before the manuscript can be accepted for publication.
- Introduction: It has a good overall quality, however a bit extensive in length. It should be shortened a bit. Also, there are some information about CI that I advise to be cited: the distribution of CI between the different clinical courses (can be found in RIS, CIS, but seems to be the most prevalent in SPMS; doi: 10.1111/bpa.12220), that it may present as a "cognitive" relapse (doi: 10.1007/s00415-011-5975-3) and that some clinical characteristics, namely male sex, lower education and higher EDSS score are predictive factors for its appearance (doi: 10.1016/j.msard.2017.06.017.)
Methods: Did the authors consider using mixed model approach for creating a regression model? The study is a follow-up study with repeated measures. Though the presented regression models do have merit, the change of the measured predictors and the predicted measure during a repetaed measure analysis is traditionally analyzed by mixed models and this may lead to new and important findings.
Results: the results are clearly presented but way too excessively. It needs to be more focused on the relevant results that were found. Every step of the regression model does not need to be reported. Also, the dichotomous variables should be defined in the methods section, not while reporting the results.
So, this whole section needs an in depth revision:
- For example: the assessment of baseline characteristics based on the BDI scores could be reported in two sentences: "We decided to test if there was any significant statistical difference in cognitive performance between patients depending on their BDI II scores across all cognitive tests performed. We found no relevant differneces between the two groups."
- some of the praragraphs could be reported in one sentence (e.g.: "Despite some variables have shown an initial correlation with baseline and 1-year MoCa scores, no predictive factor were determined by the regression analysis"), while some should be excessively shortened. Again: please, focus on the relevant results.
Discussion: should be reconsidered based on the revision of the results section.
The overall merit of the article is high, it evaluates important hot topics, but changes should be made before considered for publication.
Author Response
To our esteemed reviewer,
We thank you for your valuable feedback. We have carefully taken into account your input, and have performed the following modifications:
-introduction: we have shortened the introduction, while adding data regarding CI in different forms of MS, cognitive relapses and predictive risk factors.
-methods - When performing our statistical analysis, we consulted with an expert for choosing each performed test. Regarding the mixed model regression, we checked to see if there was a significant impact due to repeated measurements (mostly regarding sNfl values, and to a lesser degree moca-retesting at one year follow-up), but there was little to no benefit in data analysis, while creating a much more complicated model and difficulties in interpreting the results.
-results - we have shortened the results section, taking your suggestions into account.
-discussions - this section was partly rewritten to reflect the changes made to previous sections
Your observations were on point, and we believe that by addressing these issues the overall quality of the manuscript has greatly improved.
Best regards
Round 2
Reviewer 2 Report
All necessary revisions have been made and the manuscript can now be published
Author Response
Thank you for your favorable review.
Reviewer 3 Report
Unfortunately, in the revised version of the manuscript the authors failed to address sufficiently 2 of the 3 issues indicated in the previous review.
1. The authors underline that their cohort is representative of real-life MS patients but still the number of patients is very low and thus the results need to be validated in a much larger group.
2. The information regarding DMT remains very scarce. It is still unclear to what extent the observed cognitive improvement and changes in biomarkers levels were associated with DMT introduction, and thus how much the predictive model described in the study is influenced by the effects of DMTs.
In general, the study still needs to be significantly improved before publication.
Author Response
Thank you for your review.
Regarding point 1 –The authors underline that their cohort is representative of real-life MS patients but still the number of patients is very low and thus the results need to be validated in a much larger group.
We agree to your point – the number of patients is rather small, and we draw attention to this shortcoming in the Discussions segment of our article. Further, larger studies will need to confirm or contradict our findings. However, we believe the novelty of the information justifies the urgency to bring this important finding to the international research community. While many aspects are discussed, in the end, the article focuses on only one endpoint (active cognitive decline as a dichotomous variable) with sufficient final events recorded and robust statistical work included. We kindly ask you to reconsider this from the perspective that is a pilot study opening the way to larger trials, rather than anything else.
Regarding point 2: insufficient clarifications on DMT use and their impact on cognitive performance and biomarkers.
The following paragraph has been added and updated with the following information:
“Only 6 patients were started directly on high efficacy DMTs following RRMS diagnosis (either Natalizumab or Ocrelizumab), while 44 were initiated on first line agents. The median follow-up under any DMT for the study population was 10 months. The last blood sample was obtained when patients had been under DMTs for a median time of 4 months. We compared all biomarkers and cognitive tests results at on year follow-up between the 2 groups, but no statistically significant difference was found. DMTs showed no impact in predictive modelling in further analysis. Amongst the patients treated with first line agents, no significant differences were found between any of the used agents regarding cognitive performance and analyzed biomarkers. We decided not to include this variable in further analysis.”
As well as this segment in the Discussions:
“Regarding the effects of DMT use on the analyzed biomarkers and cognitive performance, no statistically significant difference was found between any of the agents used, whether high efficacy or first line agents. No predictive model showed significance or improvement when DMTs were added as a variable. This is probably due to 2 facts: the last blood samples were obtained when patients had been under DMTs for a median of 4 months, an insufficient timespan to determine significant differences between the groups. The final cognitive tests were performed when the patients had been under treatment for a median of 10 months, which again is a period too short to show an impact between different DMTs. This should be interpreted that important cognitive decline may occur in the brief timeframe until DMTs become efficient, and that early and appropriate treatment initiation is key to limiting this damage.”
Careful analysis of this data has been performed, with numerous models being tested, as well as comparisons between high, medium and low efficacy agents. Some trends can be observed, as seen in slight drops in sNfL samples, but non reach statistical significance. Due to the article already being quite long, we opted against adding all this info.
We hope that by adding the 2 paragraphs we have addressed the issue in a sufficient manner. If needed, additional materials can be shared to clear the reviewing process.
We hope our answer has addressed at least some of your concerns.
Best regards
Reviewer 4 Report
The authors reflected on all the issues I raised. I accept their answers.
Author Response
Thank you for your favorable review.